# Quality Formation of Adzuki Bean Baked: From Acrylamide to Volatiles under Microwave Heating and Drum Roasting

**DOI:** 10.3390/foods10112762

**Published:** 2021-11-10

**Authors:** Xinmiao Yao, Xianzhe Zheng, Rui Zhao, Zhebin Li, Huifang Shen, Tie Li, Zhiyong Gu, Ye Zhou, Na Xu, Aimin Shi, Qiang Wang, Shuwen Lu

**Affiliations:** 1Food Processing Research Institute, Heilongjiang Academy of Agricultural Sciences, Harbin 150086, China; cocoyococo@163.com (X.Y.); lilyamongthorns@163.com (R.Z.); lizhebin2010@163.com (Z.L.); shenhuifang_1987@126.com (H.S.); zhouye614@163.com (Y.Z.); doctorserena@163.com (N.X.); 2Institute of Food Science and Technology, Chinese Academy of Agricultural Sciences, Beijing 100193, China; shiaimin@caas.cn (A.S.); wangqiang06@caas.cn (Q.W.); 3China School of Engineering, Northeast Agricultural University, Harbin 150030, China; zhengxz@163.com; 4Crop Resources Institute, Heilongjiang Academy of Agricultural Sciences, Harbin 150086, China; haas2005@163.com; 5Gansu United Testing Standards Technical Service Co., Ltd., Lanzhou 730030, China; my_road@163.com

**Keywords:** adzuki bean, acrylamide, volatile, microwave baking, drum roasting

## Abstract

Baked adzuki beans are rich in tantalizing odor and nutritional components, such as protein, dietary fiber, vitamin B, and minerals. To analyze the final quality of baked beans, the acrylamide and volatile formation of adzuki beans were investigated under the conditions of microwave baking and drum roasting. The results indicate that the acrylamide formation in baked adzuki beans obeys the exponential growth function during the baking process, where a rapid increase in acrylamide content occurs at a critical temperature and low moisture content. The critical temperature that leads to a sudden increase in acrylamide content is 116.5 °C for the moisture content of 5.6% (*w.b.*) in microwave baking and 91.6 °C for the moisture content of 6.1% (*w.b.*) in drum roasting. The microwave-baked adzuki beans had a higher formation of the kinetics of acrylamide than that of drum-roasted beans due to the microwave volumetric heating mode. The acrylamide content in baked adzuki beans had a significant correlation with their color due to the Maillard reaction. A color difference of 11.1 and 3.6 may be introduced to evaluate the starting point of the increase in acrylamide content under microwave baking and drum roasting, respectively. Heating processes, including microwave baking and drum roasting, for adzuki beans generate characteristic volatile compounds such as furan, pyrazine, ketone, alcohols, aldehydes, esters, pyrroles, sulfocompound, phenols, and pyridine. Regarding flavor formation, beans baked via drum roasting showed better flavor quality than microwave-baked beans.

## 1. Introduction

The adzuki bean, a well-known agricultural product, has a high protein content, low fat, and diverse ingredients. The baking processes for adzuki beans may produce products with a unique flavor–texture combination. The baked powder of adzuki beans is very popular in Asian countries due to its pleasant odor, rich nutritional qualities, and ease of serving. The baked powder from whole beans contains functional ingredients in the bean’s coating, in accordance with the health concept of whole-cereal nutrition. Rotated drum roasting is a conventional method of baking treatment of adzuki beans, with a high processing capacity and the use of technology. As a new baking technology, microwave heating improves the functional properties and flavor qualities of the processed material [1] and may be used to bake adzuki beans. The principal mechanisms of microwave heating are inherent ionic conduction and dipolar relaxation of polar molecules at the rapid frequency of 2.45 GHz to generate volumetric heat inside the material within a very short time [2]. In the microwave-baking process, microwave heating causes a rapid increase in temperature and rapidly dehydrates the adzuki beans to achieve a brown color and tantalizing aroma in the final product. Compared to convection heating mechanisms, such as drum roasting, for adzuki beans, the radiant heating mechanism in microwave baking exhibits significant advantages in terms of high efficiency, easy control, and easy-to-attain equipment. However, there have been few comparisons of the two baking methods to evaluate the quality of the baked adzuki beans in terms of safety and flavor generation.

In addition, the safety of baked food has been a public concern. Heating processing may induce the production of harmful substances. Acrylamide formation in food material led to particularly high starch content when sugars and an amino acid (asparagine) reacted in starchy foods undergoing a high-temperature treatment of over 120 °C [3]. Acrylamide was detected in plant foods, such as potato products, grain products, cocoa, or coffee beans during high-temperature cooking processes, such as frying, roasting, and baking [4,5]. In laboratory studies, acrylamide, a potential carcinogen, caused cancer in animals, but at much higher acrylamide levels than those in foods [6]. The formation of acrylamide in baked food may weaken its antioxidant capacity to reduce the functional value [7]. The FDA (Food and Drug Administration) initiated acrylamide-related research on toxicology, the development of analytical methodologies, food surveys, exposure assessments, and formation migration. The formation and content of acrylamide in plant-origin material containing starch that will be heated are given more attention due to their importance for the safety and quality of food [8]. Due to the relevance of food safety, the prediction and evaluation of acrylamide content in baked food play a key role in parameter optimization and technology selection during baking processes. From the perspective of food engineering, Özge and Gökmen developed a viable approach with which to evaluate the risk factors related to acrylamide formation in cookies based on the function of acrylamide content in baking cookies with regard to temperature for a risk threshold value of 200 ppb of acrylamide [9]. Radio-frequency (RF) heating following convection heating contributes to the occurrence of excessive browning in the internal portion of baked thin biscuits with less acrylamide formation [10]. Besides temperature and components [11], the heating method, such as microwave baking or drum roasting, also dominates the formation of acrylamide in the baked material [12]. The acrylamide content of cookies processed by vacuum baking was 30% lower than in those baked the conventional way due to the low temperature in the vacuum conditions [13]. However, not enough research has evaluated acrylamide content in adzuki beans baked using microwave and roasting methods. Color may be introduced as an intuitive index to evaluate the quality of the baked product’s appearance. When baking cookies, the acrylamide content has a high correlation with the browning index, which follows the first-order reaction kinetics equation [14] and elucidates the effect of baking temperature on surface color [15]. In the Maillard reaction pathways of French fries, the glucose and fructose content directly contribute to the acrylamide formation via sugar–asparagine glycoconjugates [14]. Both the color parameter and the moisture content of the French fries have a significant correlation with the acrylamide content [5]. Therefore, the acrylamide content in baked products may be evaluated based on changes in its color.

Color, as an exterior quality index, may be used to evaluate the appearance quality of baked beans, and flavor, as an interior quality index, may characterize the smell quality of baked beans, relating to the volatile component contents. Volatile components from baked materials, for example, furan, lead to a pleasant baked aroma [16], which plays a key role in quality evaluations of bean food products, related to the optimization of baking technology parameters. The formation of and changes in the volatile components of beans are complex in baking processes [17]. Scant research has analyzed the changes in the volatile components of baked beans.

To our knowledge, little information has been published on the kinetics of acrylamide and volatile components of the formation of adzuki beans under drum-roasting and microwave-baking modes. This defect limits the quality of baked adzuki beans, and the acrylamide content cannot be controlled. Therefore, the research objectives were developed as follows.

(1)To elucidate the acrylamide formation kinetics of adzuki beans baked using microwave baking and drum roasting.(2)To investigate the changes in the volatiles and color of adzuki beans after microwave baking and drum roasting.(3)To analyze the relevance of acrylamide formation and color changes in adzuki beans during roasting.

## 2. Materials and Methods

### 2.1. Raw Material

Adzuki beans with a variety of Longyin 09-05 were collected from the planting base at Heilongjiang Academy of Agricultural Sciences (Harbin, China).

### 2.2. Microwave-Baking Processing

In the microwave-baking experiment for adzuki beans, the research focused on the formation of the kinetics of acrylamide as a function of temperature and moisture, other than the effects and optimization of microwave-baking technology. Therefore, we selected fixed-input power and beans’ mass with only one level. One hundred grams of adzuki beans were selected as experimental material and immersed in purified water for 4 h at a room temperature of 20–22 °C to achieve a moisture content of 55.04 ± 0.01% (*w.b.*). The microwave-baking experiments were conducted in a microwave workstation (MWS, FISO Technologies Inc., Quebec City, QC, Canada) and connected to a temperature sensor, as shown in Figure 1a. The immersed adzuki beans of 100 g with an initial moisture content of 55.04 ± 0.01% (*w.b.*) were placed on a glass tray inside the microwave cavity under 800 W input power. In the microwave-baking experiments, at every 1 min interval, the temperature of the experimental material was recorded, and the material was taken out to measure moisture and acrylamide content, until the baked beans achieved a mass corresponding to a moisture content below 3% (*w.b.*).

To measure the beans’ temperature during microwave heating, a fiber-optic sensor (FOT-L-SD, FISO Co., Quebec City, QC, Canada) was plugged into a single bean kernel placed in the center of the glass tray to measure the temperature. There may exist a deviation in the temperature in a single bean located in the center of a microwave workstation and the average temperature of the total beans baked. In further research, a feasible method will be developed to record the online temperature of material processed by microwave.

### 2.3. Drum-Roasting Experiments

Comparing the kinetics of acrylamide formation in adzuki beans to that under microwave baking, the input power of the roasting drum and beans’ mass was fixed at only one level. One kilogram of adzuki beans (moisture content of 12.03 ± 0.01% by weight) was roasted using an apparatus (BD-CR-D1001BB, Bideli, Guangzhou, China), as shown in Figure 1b. Experiments were carried out at atmospheric pressure under an input power of 1500 W. The experimental material’s temperature was recorded at every 1 min interval using an embodied infrared radiation thermometer in the apparatus, then taken out to measure moisture, as in Figure 1b. The roasting drum was used for the baking experiment until the baked beans achieved a mass corresponding to a moisture content below 3% (*w.b.*).

### 2.4. Color Analysis

The color was measured for whole-kernel samples using a pulsed xenon lamp tristimulus colorimeter (CR-400, Minolta, Osaka, Japan) with an 8 mm diameter test area. The instrument was standardized against a white tile, a standard fitting for the instrument, before measurements. The color was expressed in *L**, *a** and *b** scale parameters. To evaluate the overall color value in *L**, *a** and *b**, the combined expression of the color difference value ∇*E* was calculated using Equation (1).
(1)$$\nabla E=\sqrt{{\left(L^{*}-L_{0}^{*}\right)}^{2}+\left(a^{*}-a_{0}^{*}\right)+\left(b^{*}-b_{0}^{*}\right)}$$
where *L**, *a**, and *b** are the brightness value, red–green value, and yellow–blue value of the baked beans, respectively. *L*_0_*, *a*_0_*, and *b*_0_* are the brightness value, red–green value, and yellow–blue value of the raw beans, respectively.

### 2.5. Measurement of Moisture Content

A total of 3000 g of each sample was accurately weighed and then dried to a constant mass in an oven at 105 °C for 24 h, according to the Association of Official Analytical Chemists (AOAC, Gaithersburg, MD, USA) method (1995), to determine the moisture content of each sample.

### 2.6. Acrylamide Content Analysis

Acrylamide content in adzuki beans at different stages of the roasting process was determined according to the method of Bortolomeazzi, Munari, Anese, and Verardo [18], with some modifications. Fifty grams of samples was crushed by a grinder and frozen at −20 °C. Each sample was accurately weighed to 1 g (accurate to 0.001 g), and 10 μL of _13_C_3_-acrylamide (10 mg/L) was added as an internal standard working solution, followed by the addition of 10 mL ultra-pure water. Following shaking operation for 30 min, centrifugation was performed at 4000 r/m for 10 min, and the supernatant was taken for further purification.

Five milliliters of n-hexane was added to the supernatant extracted from the sample and then shaken for 10 min. The organic phase was removed following centrifugal treatment at 10,000 r/m for 5 min. The extraction was repeated with 5 mL of n-hexane, and 6 mL of aqueous phase was rapidly filtered using a 0.45 μm membrane. An HLB SPE (Solid Phase Extraction) column (5 μm 2.1 mm I.D. × 150 mm Atlantis C18 column, Waters, Milford, MA, USA) was activated using 3 mL of methanol and 3 mL of water for further experiments. A total of 5 mL of filtrate was extracted using the HLB SPE column, and the extract liquor was collected. This was followed by elution with 4 mL of methanol aqueous solution (80%). All the eluent was collected and combined with the extract liquor for purification. The Bond Elut-Accucat SPE column (3 mL, 200 mg, Agilent, Santa Clara, CA, USA) was activated using 3 mL of methanol and 3 mL of water. All the extract liquor discharged under gravity was collected, concentrated to dryness under nitrogen flow, and diluted to 1.0 mL using 0.1% formic acid solution for the test.

The information of the liquid chromatogram and analysis parameters were as follows.

Chromatograph: Xevo TQ-S Triple Quad Mass Spectrometer, Waters, Milford, MA, USA.

Chromatographic column: Atlantis C18 (5 μm, 2.1 mm I.D. × 150 mm).

Precolumn: C18 protection column (5 μm, 2.1 mm I.D. × 30 mm).

Mobile phase: methanol/0.1% formic acid (10:90, volume fraction).

Flow rate: 0.2 mL/min.

Injection volume: 25 μL.

Column temperature: 26 °C.

Mass spectrometry conditions:

Capillary voltage: 3500 V.

Cone voltage: 40 V.

RF lens 1 voltage: 30.8 V.

Ion source temperature: 80 °C.

Desolvation temperature: 300 °C.

Ion collision energy: 6 eV.

Acrylamide: parent ion *m*/*z* 72, daughter ion *m*/*z* 55, daughter ion *m*/*z* 44.

_13_C_3_ acrylamide: parent ion *m*/*z* 75, daughter ion *m*/*z* 58, daughter ion *m*/*z* 45.

Quantification ion: *m*/*z* 55 for acrylamide, *m*/*z* 58 for _13_C_3_ acrylamide.

The method of signal-to-noise ratio (SNR) was used to determine the quantification limit (LOQ) and the detection limit (LOD). The concentration corresponding to 10 times the SNR was used as the LOQ, and the concentration corresponding to 3 times the SNR was used as the LOD, represented by Equations (2) and (3).
LOQ = 10 N/S(2)
LOD = 3 N/S(3)
where N is noise, and S is detector sensitivity.

The LOQ of the method is 10 μg/kg, and the absolute difference between the two independent measurements obtained under repeatability conditions should not exceed 20% of the arithmetic mean.

### 2.7. Volatile Components Measurement

A purge and trap sampling device (homemade, tailored to experiments) was used to detect volatile components of the baked beans, as shown in Figure 1c. A 20 g baked bean sample was placed into the sampling jar with a volume of 250 mL. Considering the boiling point of volatile components, the sample was heated using an air flow of 40 mL/min under a temperature of 40–120 °C with 20 °C intervals. The purge and trap sampling duration was set as 60 min to collect volatiles at a volume of 2.4 L.

The analysis of volatile substances was carried out by concentrating volatile compounds using the purging and trapping method, then introducing volatile compounds into GC-MS (Gas Chromatography-Mass Spectrometer) using a thermal desorption supplementary collector for qualitative and quantitative analysis. This method focuses on the relative strength of these volatiles, rather than the absolute composition of flavor-producing substances. Therefore, two quantitative methods were used: one was the normalization method, and the second quantified all peak substances using the relative correction factor of toluene, which represents the relative content of the volatiles collected using this test method. The response factor of toluene (F) was obtained by the standard curve method. The concentration of volatiles (C) in baked beans was determined by Equations (4) and (5).
(4)$$F=\frac{\text{Peak}\;\text{area}\;\text{of}\;\text{toluene}}{\text{Toluene}\;\text{content}\;\text{in}\;\text{standard}\;\text{curve}}$$
(5)$$C\left(\text{Volatile}\right)=\frac{\text{Peak}\;\text{area}\;\text{of}\;\text{volatile}\times F}{\text{Mass}\;\text{of}\;\text{sample}}\times \text{Unit}\;\text{conversion}\;\text{factor}$$
where the mass of sample was 20 g in Equation (5).

The information of gas chromatogram and analysis parameters were as follows.

Chromatograph: Unity-xr/7890B-5977B Thermal Desorption Gas Chromatography-Mass Spectrometry, Agilent, Santa Clara, CA, USA.

Chromatographic column: DB-5ms 60 m × 0.25 mm × 0.25 μm, Agilent, Santa Clara, CA, USA.

Initial temperature: 40 °C, maintain 5 min; 3 °C/min to 160 °C, maintain 2 min; 8 °C/min to 260 °C, maintain 2 min.

Carrier gas: helium.

Flow rate: 1 mL/min.

Mass spectrometry:

Electron energy: 70 eV.

Ion source temperature: 230 °C.

Quadrupole temperature: 180 °C.

Transmission line temperature: 230 °C.

Scan range: 31 *m*/*z* to 500 *m*/*z*.

Thermal desorption temperature: 280 °C for 5 min.

Filling temperature of cold trap: −15 °C.

Analytical temperature of cold trap: 300 °C for 5 min.

### 2.8. Model Development

According to the acrylamide changes in beans processed by microwave baking and drum roasting, the SigmaPlot software (Version 12.5, Systat Software, Inc., San Jose, CA, USA) method was employed to fit the power function. In the preliminary regression treatment of experimental data, it was observed that the change trends of acrylamide in adzuki bean baked had high consistency with the curves of the power function. According to the data of acrylamide content in adzuki beans with baking duration (t), temperature (T), moisture content (M), and ratio of temperature to moisture content (T/M) under microwave baking and drum roasting, the power model types from the SigmaPlot software (Version 12.5, Systat Software, Inc.) were selected to be fitted by using the iterative regression method to constant and kinetic coefficients. The determined coefficient (R^2^) and relative square error (RSE) were introduced to evaluate the fitting models. The closer to 1 the R^2^ value, the less the RSE indicates the more reasonable fitting model. The developed power models characterize the formation kinetics of acrylamide. The function characterizes the formation kinetics of acrylamide. The determined coefficient and relative error were introduced to evaluate the fitting models.

### 2.9. Statistical Analysis

Three repeated experiments were conducted for every operation, and average data were expressed by mean ± standard deviation (SD). Three repetitions were performed in individual determination such as analysis of dry matter, acrylamide content, and composition of volatile compounds. The data were statistically processed by using analysis of variance (ANOVA) and Tukey’s honestly significant difference (HSD) test, which were performed using the SPSS statistics software (Version 19.0, SPSS Inc., Chicago, IL, USA). Statistically significant differences were tested at a 5% probability level (*p* < 0.05).

## 3. Results and Discussion

### 3.1. Formation Kinetic of Acrylamide in Adzuki Beans under Microwave Baking and Drum Roasting

Adzuki beans are abundant in starch, which easily forms acrylamide during heating treatments as a result of the Maillard reaction, as the main chain from asparaginase in amino acids forms acrylamide molecules [19]. The reducing sugar reacts with free amino groups of amino acids or protein to produce the Maillard reaction in baking processes with a high temperature [20].

#### 3.1.1. Changes in the Acrylamide Content of Adzuki Beans during Microwave Baking

The change in the acrylamide content of adzuki beans during microwave heating is presented in Figure 2a.

As shown in Figure 2a, no acrylamide was detected inside adzuki beans until after 4 min of microwave baking; then, acrylamide gradually formed, reaching 32.8 μg/kg at 9 min, followed by a rapid increase to 432.9 μg/kg. These changes are attributed to the three stages of the Maillard reaction, as follows [21]. In Stage 1, with the absorption of the thermal energy into baked adzuki beans, the amino acids react by reducing sugar to form Amadori molecular product rearrangement (APR). In this stage, the carbonyl group of reducing sugars and the amino of amino acids form Schiff with a C=N bond. As a high-activity medium substance, the rearrangement of Schiff generates the APR. In Stage 2, with the further accumulation of thermal energy inside baked adzuki beans, the degradation of APR generates hundreds of volatile substances. In this stage, the rising temperature causes the degradation of amino to form carbonyl substances during the Maillard reaction, including aldehydes, ammonia, and ribose, under the Strecker degradation mechanism. In Stage 3, when the thermal accumulation achieves the active energy of the Maillard reaction at the critical temperature of acrylamide formation, asparagine and carbonyl compounds, as the precursor substances, rapidly synthesize the acrylamide inside the baked material.

The change in the acrylamide content of adzuki beans under microwave baking conditions followed the exponential growth function with a kinetic constant of 1.25 (1/s), embodied in Figure 2a. Acrylamide content is at low levels until a critical point, which provides guidance for controlling the acrylamide content of adzuki beans under microwave baking.

In the baking processes of adzuki beans, the formation of acrylamide is an endothermic reaction, where the reaction rate constant, as a function of *T* and *M*, is characterized by a modified Arrhenius equation (Equation (6)).
(6)$$k=k_{ref}\times \text{exp}\left[\frac{-E_{a}}{R}\times \left(\frac{1}{T}-\frac{1}{T_{0}}\right)\right]\times \text{exp}\left[\frac{-H_{a}}{R\times T}\times \left(M-M_{0}\right)\right]$$
where *k* is the reaction rate (1/s), min^−1^; *k_ref_* is the constant of the reaction rate (1/s); *E_a_* is the reaction active energy, kJ/mol; *R* is the gas constant, 8.314 J/(mol·K); *T* is the temperature, °C; *T*_0_ is the initial temperature of bean processed at room temperature, °C (20 °C); *H_a_* is the reaction heating, kJ/kg; *M* is the moisture content (%); and *M*_0_ is the initial moisture content (*w.b.*, %).

From Equation (1), the kinetics of acrylamide formation in adzuki beans are shown to depend on the changes in temperature and moisture content. As shown in Figure 2b, microwave volumetric heating causes the temperature to rise with the reduction in adzuki beans’ moisture content. Both temperature and moisture content influence the chemical reaction of acrylamide formation, which was presented using the temperature to moisture ratio (*T*/*M*) as an exponential growth function, as shown in Equation (7).
(7)$$y_{MR}=6.94e^{0.05\times \frac{T}{M}},R_{2}=0.9985,\text{SEE}=6.78$$
where *y_MR_* is the acrylamide content inside the baked beans considering the ratios of temperature (*T*, °C) and moisture content (*M*, %, *w.b.*). *R*^2^ is the determinate coefficient, and *SEE* is the sum of squared residuals. When adzuki beans were baked in a microwave, the acrylamide resultant was formed with increasing temperature and decreasing moisture content. The effect of moisture on the acrylamide formation of adzuki beans obeys the exponential rise to maximum function shown in Equation (7), and the effect of temperature on the acrylamide formation obeys the exponential growth function, as shown in Equation (8).
(8)$$y_{MM}=1602.68-1598.74\left(1-e^{-0.79M}\right),R_{2}=0.9990,\text{SEE}=6.31$$
(9)$$y_{MT}=e^{0.29\left(T\;-\;116.51\right)},R_{2}=0.9979,\text{SEE}=8.01$$
where *y_MM_* is the acrylamide content in beans baked by microwave heating considering the moisture content (*M*, *w.b.*, %), and *y_MT_* is the acrylamide content in beans baked by microwave heating considering temperature (*T*, °C). The rate constant *k* of acrylamide formation in Equation (1) has a positive correlation with the temperature of adzuki beans due to the high temperatures needed for acrylamide formation.

According to the results of Equation (9), the critical temperature needed for acrylamide content was at 116.51 °C, corresponding to a moisture content of 5.6% (*w.b.*). Therefore, microwave baking may produce baked beans with no detectable acrylamide content when the heating temperature is below 116 °C, with a moisture content higher than 5.6%.

#### 3.1.2. Changes in the Acrylamide Content of Adzuki Beans during Drum Roasting

The changes in the acrylamide content, temperature, and moisture content of adzuki beans are presented in Figure 3a,b.

In drum roasting, no acrylamide content is detectable in adzuki beans for 7.5 min; then, there is a clear rising trend up to 87.4 μg/kg for 23 min, as shown in Figure 3a. The change in acrylamide content in adzuki beans under drum-roasting conditions obeys the exponential growth function with a kinetic constant of 0.1225 (1/s), as shown in Figure 3a.

In the drum-roasting processes, the acrylamide content of adzuki beans as a function of the temperature to moisture content ratio (shown in Figure 3b) obeys the exponential growth function, as shown in Equation (10).
(10)$$y_{D}=2.44e^{0.13\times \frac{T}{M}},R_{2}=0.8834,\text{SEE}=10.42$$
where *y_D_* is the kinetic constant of the acrylamide content of adzuki beans, considering the temperature to moisture ratio; *T* is the temperature of the baked beans (°C); and *M* is the moisture content (%, *w.b.*).

The temperature to moisture content ratio in drum roasting (0.13) in Equation (10) is higher than that in microwave baking (0.05) in Equation (7). However, the kinetic coefficients of moisture in Equation (11) and temperature in Equation (12) for drum-roasted adzuki beans are lower than those of microwave-baked beans, as shown in Equations (8) and (9).
(11)$$y_{DM}=926.84-951.42\left(1-e^{-0.36M}\right)\;,R_{2}=0.9305,\text{SEE}=8.48$$
(12)$$y_{DT}=e^{0.07\left(T-91.62\right)},R_{2}=0.9796,\text{SEE}=4.36$$
where *y_DM_* is the acrylamide content in drum-roasted beans, considering moisture content (*M*, %), and *y**_DT_* is the acrylamide content in drum-roasted beans, considering temperature (*T*, °C). According to Equation (8), the critical temperature causing acrylamide content to rise in adzuki beans during drum roasting is 91.62 °C at a corresponding moisture content of 6.1% (*w.b.*).

In summary, the kinetics of acrylamide formation in adzuki beans during microwave baking are higher than those found during drum roasting. The difference in the kinetic behavior of the acrylamide formation in adzuki beans is attributed to the greater heating intensity of microwave volumetric heating compared to drum roasting. The greater heating intensity results in higher temperatures and a quicker reduction in moisture content in adzuki beans baked using microwave volumetric heating. When baking adzuki beans using a microwave, the microwave field causes the rapid rotation of polar molecules, at 2.45 × 10^9^ times per second. The rapid movements of polar molecules cause fierce friction and collision, producing volumetric heating, and increasing the entropy value. The increment of entropy inside microwave-baked beans may reduce the activation energy. Therefore, the low kinetic coefficient of acrylamide formation in baked beans is due to the reduction in the activation energy of precursor substance molecules, including asparagine and carbonyl compounds. In drum-roasted adzuki beans, material is heated from the surface, and inside via the convection transfer route. Compared to microwave baking, the low heating intensity in drum roasting results in a slowly increasing temperature and decreasing moisture content in the baked materials, as the precursor substance molecules need more activation energy to form acrylamide. Therefore, the acrylamide content in adzuki beans remains at low levels until high temperatures (116 °C) are reached during microwave baking, which improves the baking efficiency. However, the initial temperature of acrylamide formation under drum roasting is 91.6 °C, which is lower than the consensus temperature of 120 °C [22]. If baking processes have a long-enough duration, high temperature, and low moisture content, they contain the conditions for acrylamide formation.

### 3.2. Changes in the Color of Adzuki Beans during Microwave Baking and Drum Roasting

As shown in Figure 4a,b, the *L**** value of adzuki beans tends to decrease, but the *a** and *b** values show fluctuating changes due to the formation of Amadori intermediates and Maillard reaction products during microwave baking and drum roasting.

In the microwave-baking process, the color changes obey the exponential decay with two stages, as shown in Equation (13).


(13)
$$\nabla E=16.47e^{{\left(-2.95E-11\right)}^{t}}-15.38e^{-0.11t}$$


From Equation (13), the overall color of adzuki beans tended to slow, then showed a sharp decline due to the thermal accumulation inside baked beans resulting from microwave volumetric heating (*p* < 0.05), which caused an obvious Maillard reaction in adzuki beans in a short time period.

In the drum-roasting processes, the color changes obey the same rule as those in microwave baking, as shown in Equation (14).


(14)
$$\nabla E=0.73e^{0.08t}+1.24e^{0.08t}$$


From Equation (14), the overall color of adzuki beans shows a gentle decline at two stages due to the heat diffusion of convection and conduction inside baked adzuki beans.

The acrylamide formation of baked adzuki beans has a high correlation with its color level due to the Maillard reaction. Although no statistically significant correlation was established for acrylamide content with a single color value of *L****, *a****, and *b****, the second term in Equations (13) and (14) presents the changes in the color of adzuki beans as acrylamide content rapidly rose under microwave-baking and drum-roasting technologies, respectively. According to the critical duration of 9.5 min and 7.5 min in Figure 2a and Figure 3a, presenting a rapidly rising acrylamide content (*p* < 0.05), ∇*E* was determined from Equations (13) and (14) as 11.1 and 3.6 under microwave baking and drum roasting, respectively. There is reason to infer that the initial point of the formation of acrylamide content in baked beans may be evaluated based on its overall color ∇*E* level under different baking methods, although further quantitative validation is needed.

### 3.3. Changes in Volatile Compounds in Baked Bean in Baking Processes

The chromatograms of baked beans, measured by the purge and trap sampling method, are shown in Figure 5 under airflow temperatures of 40 °C, 60 °C, 80 °C, 100 °C, and 120 °C. More volatile components were collected under higher airflow temperatures.

The selected measurement temperature was 100 °C, referring to the brewing temperature of adzuki bean powder with boiling water. Based on the species and concentration of volatiles from baked adzuki beans, shown in Figure 5 at 100 °C, we classified and added the test results to find the changes in the characteristic curves of volatile compounds in baked beans during the microwave-baking (Figure 6*a*–*e*) and drum-roasting processes (Figure 6*a*’–*e*’). The data in detailed in Figure 6 were presented in Appendix A.

According to the GC-MS results for the baked beans, as shown in Figure 6, an average of 48 volatile components are detected in microwave baking, and an average of 56 volatile components are detected after drum roasting. Characteristic volatile compounds in baked beans are determined as furan, pyrazine, ketone, alcohols, aldehydes, esters, pyrroles, sulfocompound, phenols, and pyridine. The precursor of these volatile substances is produced during the thermal treatment of baked beans with the formation of organic acid, which results from the thermal degradation of polysaccharide to produce the acidity. The thermal decomposition of oligosaccharide in beans generates the reducing sugar. The Maillard reaction between the reducing sugar and free amino acid produces volatile substances, including pyrazines, furans, and aldehydes [23]. Further high-temperature baking, and the stark degradation reaction between α-amino acid and dicarbonyl compounds, form aldehydes and ketone. These characteristic compounds present odor as a cream from aldehyde, delicate fragrance from alcohols, fruit odor from esters, and sour from acids. In baking processes, including microwave and convective heating, high temperatures result in the production of characteristic volatile compounds in baked beans. In a general trend, the maximum contents of volatile compounds were higher in drum roasting than that in microwave baking. As shown in Figure 6*a*,*a*’, in microwave baking and drum roasting, the formation of furan led to the pleasant smell of baked beans. With different bean-baking processes, a high temperature resulted in a clear reduction in furan. Excessive baking with high temperatures and low moisture contents caused the fine volatiles to almost disappear. The aldehyde content of baked beans has similar trends as the furnace changes in microwave and drum roasting baking, as shown in Figure 6*b*,*b*’. The alcohol content increases then decreases with the microwave- and drum-roasting processes shown in Figure 6*c*,*c*’. Figure 6*d*,*d*’ shows that the ester content of baked beans increases in microwave baking and first increases then decreases in drum roasting. From Figure 6*e*,*e*’, drum roasting leads to the formation of pyrazine, an attractive volatile with a cocoa aroma, and microwave baking is not good for the formation of pyrazine. Although little research has been conducted on the formation of acrylamide content and volatile compounds in baked beans, in the existing research, cocoa bean roasting increased the content of acrylamide 2–3-fold, which indicates that roasting cocoa beans increases the content of furan by 25.1–34.8 ng g^−1^ [4]. Milling baked adzuki beans may be used as the raw material for instant food, as shown in Figure 7. This demonstrates the importance and novelty of quality formation and control based on baking methods and technology parameters for baked adzuki beans.

## 4. Conclusions

The kinetics of acrylamide formation in adzuki beans under microwave baking are higher than those under drum roasting due to the greater kinetic coefficient in microwave baking. The critical temperature leading to this increase in acrylamide content is 116.5 °C, with a corresponding moisture content of 5.6% (*w.b.*) in microwave baking, and 91.6 °C, with a corresponding moisture content of 6.1% (*w.b.*). The starting point indicating the increase in acrylamide content is ∇*E* of 11.1, 3.6 under microwave baking and drum roasting, respectively. No statistically significant correlation was found for acrylamide content with a single color value of *L**, *a**, and *b**; however, the overall color of adzuki beans tends to decline with a rapidly rising acrylamide content, and the initial point of the formation of acrylamide content in baked beans may be evaluated based on its overall color ∇*E* level under the microwave-baking and drum-roasting methods. Characteristic volatile compounds in baked beans include furan, pyrazine, ketone, alcohols, aldehydes, esters, pyrroles, sulfocompound, phenols, and pyridine. The comparison of volatile formation in beans baked using drum roasting and microwave baking needs further investigation to evaluate flavor quality. This can provide guidance for controlling the acrylamide content in adzuki beans under microwave baking and drum roasting.

## Figures and Tables

**Figure 1 foods-10-02762-f001:**
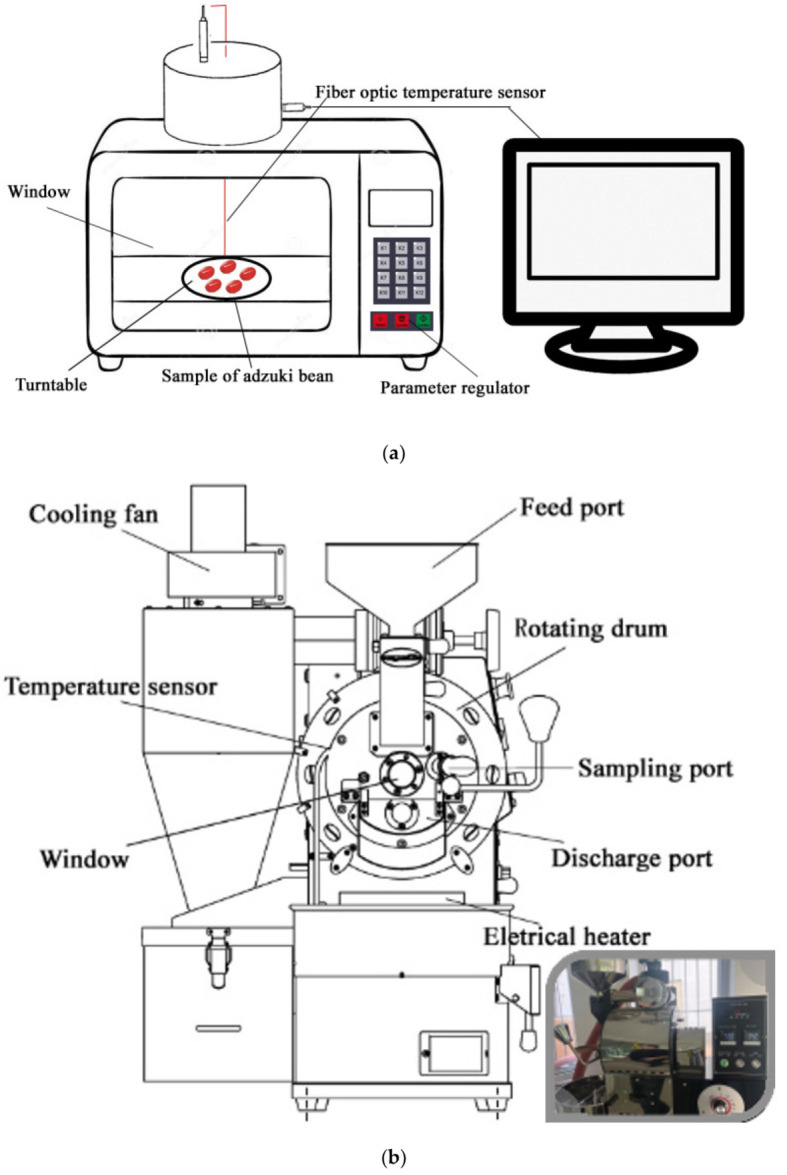
(**a**) Diagram of microwave workstation for baking treatment of adzuki beans. (**b**) Roasting drum device for baking treatment of adzuki beans. (**c**) Purge and trap device for detection of volatile compounds in baked beans.

**Figure 2 foods-10-02762-f002:**
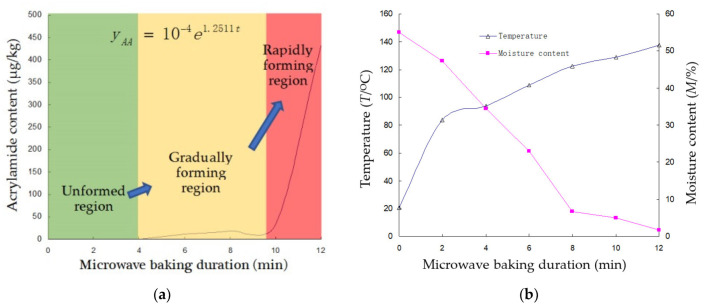
(**a**) Acrylamide content of adzuki beans under microwave baking. (**b**) Temperature and moisture content of adzuki beans after microwave baking.

**Figure 3 foods-10-02762-f003:**
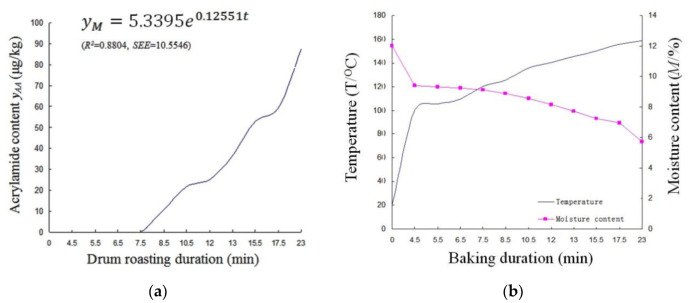
(**a**) Acrylamide content of adzuki beans during drum roasting. (**b**) Temperature and moisture content of adzuki beans with drum roasting.

**Figure 4 foods-10-02762-f004:**
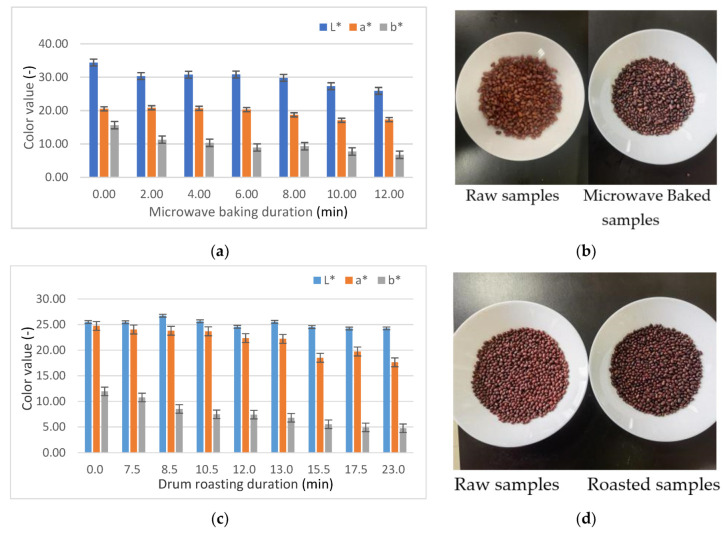
(**a**,**b**) Color changes in adzuki beans during microwave baking. (**c**,**d**) Color changes in adzuki beans during drum roasting.

**Figure 5 foods-10-02762-f005:**
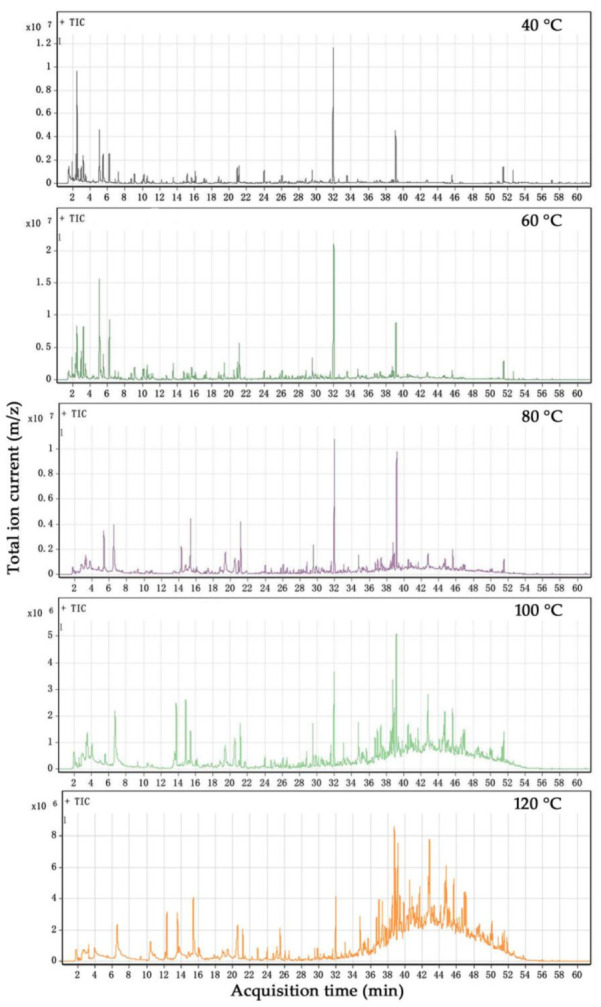
Total ion chromatogram (TIC) of baked beans under different temperatures.

**Figure 6 foods-10-02762-f006:**
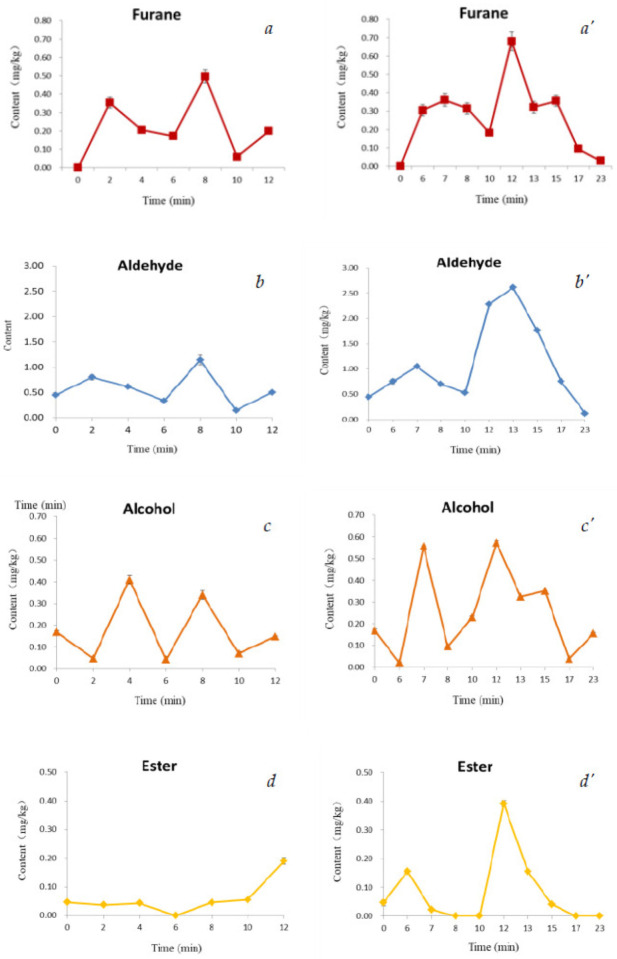
Changes in characteristic volatile compounds of baked beans during microwave-baking (***a***–***e***) and drum-roasting processes (***a*’**–***e*’**).

**Figure 7 foods-10-02762-f007:**
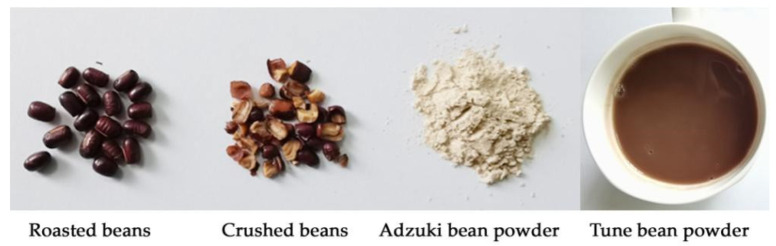
Baked adzuki bean, powder, and tune bean drinks.

## Data Availability

The data presented in this study are available on request from the corresponding author.

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
