# Peer review of "Quality Formation of Adzuki Bean Baked: From Acrylamide to Volatiles under Microwave Heating and Drum Roasting"

_foods, 2021, doi:10.3390/foods10112762_

Round 1

Reviewer 1 Report

The article titled: "Quality formation of adzuki bean baked: from acrylamide to flavor under microwave heating and drum roasting" submitted to Foods, presents very interesting and novel topic related to adzuki bean. Experiments are properly designed,  quality of paper is fine. I would like ask authors to explain which nutritional components are present in Adzuki bean, as authors stated in first sentence of abstract: " Adzuki bean baked is rich in tantalizing flavor and nutritional components". Additionally please provide name and company of instruments you used (ex. Purge & Trap sampling device is missing) or please state that it was home-made tailored to experiments. 

Author Response

Point 1: The article titled: "Quality formation of adzuki bean baked: from acrylamide to flavor under microwave heating and drum roasting" submitted to Foods, presents very interesting and novel topic related to adzuki bean. Experiments are properly designed,  quality of paper is fine. I would like ask authors to explain which nutritional components are present in Adzuki bean, as authors stated in first sentence of abstract: " Adzuki bean baked is rich in tantalizing flavor and nutritional components".

Response 1: Thanks reviewer positive comments.  According to reviewer`s suggestion, we modified first sentence of abstract as “Adzuki bean baked is rich in tantalizing volatiles and nutritional components, such as such as protein, dietary fiber, vitamin B and minerals.”.

Point 2:Additionally please provide name and company of instruments you used (ex. Purge & Trap sampling device is missing) or please state that it was home-made tailored to experiments. 

Response 2: We checked and added the instruments information in revised manuscript in line 164.

Reviewer 2 Report

After reading the article, I noticed it lacks key methodological information, the presentation of results is of poor quality (not scientific described except for kinetics), and there is no discussion of the literature. The quality of the graphs is poor. Article needs language correction in many places.

I provide comments below for future use:

line 53-54, line 91-92 – please add to the list also cocoa beans and chocolate according to the article: https://dx.doi.org/10.1021/acs.jafc.0c00412 These authors indicate in the abstract that: Roasting of cocoa beans increased the content of acrylamide 2−3-fold. The obtained results indicate that acrylamide might be formed during wet conching. Roasting of cocoa beans increased the content of furan from <LOD to25.1−34.8 ng g−1.

line 65 - the authors cite Özge and Gökmen while authors of this paper is Açar & Gökmen - please fix this error

Materials and methods

line 112 - adzuki beans were immersed in what, how long? - give details

line 113- was put on the loading tray

line 114-115- “In microwave baking experiments, the experimental material was taken out to measure temperature, moisture and acrylamide content at setting duration.” – Here you must give details about setting duration - at what intervals samples were taken for analysis?, in how many replicates was the experiment conducted? Did the sensor automatically give the temperature? Because from the description, it seems that the temperature was measured in the samples taken out but they may have cooled during removal. Please explain/clarify.

line 118- “In microwave baking, the microwave input power was set at 800 W.” – you have repeated the information previously given in line 114. Please delete this sentence.

Add subsection about temperature measurements, give details about measurement procedure, give details about the model and manufacturer of used fiber optic temperature sensor, and state the principle of its functioning.

line 125-127 – give details about embodied sensor; clear sentence from Fig 1b caption

Give details about temperature of drum roasting process, at what intervals samples were taken for analysis? in how many replicates was the experiment conducted?

Color analysis – please give information about standard observer, and an illuminant of CR-400 colorimeter. In how many replicates was the measurement taken?

Measurement of moisture content – Please state in how many replicates was the measurement taken.

Acrylamide content analysis – please describe how samples were extracted by water (purity, proportion to sample), how long, on what, what temperature? The samples were surely crushed and homogenized - please describe this procedure as well.

Was the SPE column commercially available or was its composition composed by the authors? Please provide model name and company or provide names and suppliers of individual sorbents of the column. Give purification procedure of sample on SPE column.

Give details of liquid chromatography-tandem mass spectrometry analysis. Provide LOD and LOQ of the method. Provide model and manufacturer of instrument. How relative response factor was calculated ?

Figure 1b and 1 c – provide larger images - current ones are unreadable

Flavor components measurement- Instead of the word flavour, the authors should use the word volatiles throughout the article. Whether the beans were crushed or placed into device as whole beans? How the volatile profile was analyzed and on what?

Please made subsection with informations about formation kinetic calculations.

Results and discussion

3.1. Analysis of acrylamide formation of adzuki beans in microwave baking and drum roasting  - please delete this subsection - this is known knowledge and obvious observations, already mentioned in introduction section

Please provide better quality of figure 2a - axis titles are separate from the graph and the graph itself barely shows the curve. Mark also on the graph the three stages of Millard reaction mentioned in the text.

Please provide better quality of figure 2b - axis titles are separate from the graph and the graph itself barely shows the curve.

line 225-226 - how the authors arrived at these formulas?

lines 234-236 - What was the LOD and LOQ of the method? In my opinion, the conclusion is too far-fetched. The processing time at a given temperature is also important here. Acrylamide will form when treated below 110 °C at 6% humidity, but after a longer time. The authors themselves have stated that at lines 291-292.

Please provide better quality of figure 3b - axis titles are separate from the graph and the graph itself barely shows the curve, no legend of curves are visible.

line 263-264 - how the authors arrived at these formulas?

How the authors have accounted for differences in initial bean moisture content during microwave and drum roasting treatment? Initial humidity certainly had an effect on acrylamide formation.

line 294-296- The authors give arbitrary conclusions even before discussing the results. It looks like they are proving their own theory. Although they write below that no significantly statistical correlation was found of acrylamide content with single color value of L, a, b.

line 296-297- “color value of adzuki bean tend to decrease” - which value?

line 297-303 – this fragment should be put to the methods section

Figure 4 a, b - The standard deviation of the L-value is the same on each bar. The same situation happens for the a-value and the b-value.

Shouldn't parameters a* and b* increase due to the formation of Amadori intermediates and Maillard reaction products? The authors show the opposite trend.

Figure 5- The temperature given in Figure 5 and its title suggest that the roasting temperatures were 40, 60, 80, 100, 120. Meanwhile, these temperatures refer to air-flow Purge & Trap sampling method.

line 340-341- The sentence needs language correction. Please provide a table of identified volatile compounds.

Figure 6- How the authors were able to quantify the concentration of the different groups of compounds that make up the odor profile. Please provide information at method section.

line 342-343- “Detected flavor components of baked bean tend to the decreasing trend at the end of baking processing.” This sentence is not a scientific description of the results. There is no summary of the quantitative percentage of volatile compounds. Please remove it.

Whole description of changes of volatile compounds in baked bean during processing is not on a scientific level. It needs rewriting.

Conclusions

line 373-374 – Please explain what does mean that: “The kinetics of acrylamide formation in adzuki beans under and microwave baking is higher than that under drum roasting.”

line 378-379- “In microwave baking, 21 flavor components are 378 detected and 23 flour components in heating drum baking.” – this is not a conclusion

The authors did not report conclusions regarding color and relationship to acrylamide content.

lines 381-382 – “In the view of favor formation, baked bean by drum roasting has the better favor quality than that by microwave baking.” - On what basis this conclusion was made - no olfactometric testing was done.

Reviewer 3 Report

The subject of the study is interesting and the manuscript brings a nice addition to the field of food processing. The authors investigated the acrylamide formation kinetics of Adzuki beans after microwave baking and drum roasting. The changes of the color and flavor compounds were also investigated during the baking processes. The text gives the impression that it has been written fast and therefore not properly checked. The paper has a number of typographical and grammatical errors which must be corrected. Some sections in the experimental part have to be written again to clarify the design of the experiments. The authors should emphasize the scientific novelty and importance of the study. The obtained results should be compared with the literature data. In the discussion the authors shall give a correlation between acrylamide content, color and flavor compounds. 

Author Response

Point 1: The subject of the study is interesting and the manuscript brings a nice addition to the field of food processing. The authors investigated the acrylamide formation kinetics of Adzuki beans after microwave baking and drum roasting. The changes of the color and flavor compounds were also investigated during the baking processes.

R: Thanks for reviewer`s positive comments.

Point 2: The text gives the impression that it has been written fast and therefore not properly checked. The paper has a number of typographical and grammatical errors which must be corrected.

R: Sorry about these mistakes in initial manuscript. We checked whole manuscript and modified typographical and grammatical errors through manuscript.

Point 3: Some sections in the experimental part have to be written again to clarify the design of the experiments. The authors should emphasize the scientific novelty and importance of the study.

R: Sorry about these deficiency in initial manuscript. We rewrote the experimental part to develop clearly experimental route as shown in revised manuscript at section 2.2 and 2.3.

Point 4: The obtained results should be compared with the literature data.

R: Thanks for your suggestion. We added the reference in revised manuscript as  Kruszewski Bartosz * and MieczysÅ‚aw WiesÅ‚aw ObiedziÅ„ski. Impact of Raw Materials and Production Processes on Furan and Acrylamide Contents in Dark Chocolate.  J. Agric. Food Chem. 2020, 68, 8, 2562–2569.

Roasting of cocoa beans increased the content of acrylamide 2−3-fold, which indicates that the roasting of cocoa beans increased the content of furan in 25.1−34.8 ng g−1 as shown in line 411 to 416 in revised manuscript.

Point 5: In the discussion the authors shall give a correlation between acrylamide content, color and flavor compounds.

R: Although little research about the formation of acrylamide content and volatile compounds in baked bean, in existing research, roasting of cocoa beans increased the content of acrylamide 2−3-fold, which indicates that the roasting of cocoa beans increased the content of furan in 25.1−34.8 ng.g−1. This hit the importance and novelty of quality formation and control based on baking method and technology parameters for adzuki bean baked as shown in line 424 to 428 in revised manuscript.

Round 2

Reviewer 2 Report

The authors have made many improvements but not all. Some comments have been omitted or commented on in general terms. Therefore, I recommend sending the article for major revisions. Here I give most important comments and holes in manuscript to be filled. Additionally, in some places the article still needs major language corrections.

line 55 – please add to the plant list also cocoa beans

line 112-113- authors did not provide the liquid in which the adzuki beans were immersed

line 117-120 – Authors did not responded to comment: in how many replicates was the experiment conducted?

line 124- Was the measurement taken only on a single bean?

acrylamide determination – effluent in SPE extraction? Please verify the word.

line 167- provide name of supplier of Bond Elut-Accucat SPE column

line 171 - How was the LOQ calculated? and what about LOD? give formulas, procedure of  determination of LOD and LOQ

line 171-172 – model TQS of Waters but which one? provide informations about chromatograph and detector. Please provide the analysis parameters of determination and solvents used for the mobile phase. State on which chromatography column the analysis of acrylamide in the chromatograph was performed?

Volatile components measurement – provide GC-MS model and manufacturer names, provide the provide the analysis parameters of determination. State on which chromatography column the analysis of volatiles was performed?

line 186- odors? better volatiles

subsection 2.8 - contains scarce and general information. Please give details of the procedure for calculating kinetics.

line 195- in formula you use quality of the sample? what are the values?

line 203- two information is needed:  1) in how many replicates was drum roasting performed and in how many was microwave heating performed;  2) in how many repetitions were performed individual determinations such as analysis of dry matter, acrylamide content, composition of volatile compounds

The tables of identified volatile compounds under microwave baking and drum roasting which was provided in cover letter should be formatted and provided with article as supplementary material.
